# Regeneration and Genetic Transformation in *Eucalyptus* Species, Current Research and Future Perspectives

**DOI:** 10.3390/plants13202843

**Published:** 2024-10-11

**Authors:** Youshuang Wang, Zhihua Wu, Xiaoming Li, Xiuhua Shang

**Affiliations:** 1Research Institute of Fast-Growing Trees, Chinese Academy of Forestry, Zhanjiang 524022, China; wangyoushuangjy@163.com (Y.W.); wzhua2889@163.com (Z.W.); lixiaoming203@163.com (X.L.); 2State Key Laboratory of Efficient Production of Forest Resources, Beijing 100083, China

**Keywords:** *Eucalyptus*, plant regeneration, genetic transformation, organogenesis, somatic embryogenesis

## Abstract

*Eucalyptus* is an important plantation tree with a high economic value in China. The tree contributes significantly to China’s timber production. The stable and efficient *Eucalyptus* regeneration system and genetic transformation system are of great significance for exploring the regulatory function and possible genetic breeding capacity of important genes in the species. However, as a woody plant, *Eucalyptus* has problems, such as a long generation cycle, strong specificity of the regeneration system, and a low genetic conversion rate, which seriously limit the rapid development of *Eucalyptus* genetics and breeding programs. The present review summarizes the status of research on *Eucalyptus* regeneration and genetic transformation, with a focus on the effects of explants, media, plant growth regulators (PGRs), and concentrations in the *Eucalyptus* regeneration process. In addition, the effects of genotype, *Agrobacterium*, antibiotics, preculture, and co-culture on the genetic transformation efficiency of *Eucalyptus* are discussed. Furthermore, the study also summarizes the problems encountered in *Eucalyptus* regeneration and genetic transformation, with reference to previous studies, and it outlines future developments and prospects. The aim was to provide a reference for solving the problems of genetic instability and the low transformation efficiency of eucalyptus, and to establish an efficient and stable eucalyptus regeneration and transformation system to accelerate the process of its genetic improvement.

## 1. Introduction

*Eucalyptus* belongs to the Myrtaceae family, which includes *Eucalyptus*, *Angophora*, and *Corymbia*. *Eucalyptus* mainly originates from Australia, Tasmania, and the surrounding islands, and is one of the world’s most famous rapid-growing tree species [1]. Eucalyptus grows in regions with various climatic conditions and soils, ranging from low tropical regions to high-altitude regions. The naturally occurring eucalyptus forest area worldwide is over 40 million hectares [2]. In addition, since the 19th century, *Eucalyptus* plantations have been artificially introduced to over 100 nations and regions across the world [3]. China has a *Eucalyptus* plantation history of over 130 years, with the early usage of *Eucalyptus* mostly involving courtyard greening and the sporadic planting of tree species. In the 20th century, broad-leaved trees and cultivation research cooperation projects were introduced in China and Australia in the 80s, after which the *Eucalyptus* introduction and domestication in China witnessed a breakthrough in progress. The cold-resistant *Eucalyptus* tree species, such as *E. globulus* Labill., *E. maidenii* F. Muell. , *E. nitida* Hook. f., and *E. dunnii* Maiden, have been screened out, along with the establishment of breeding hybrid clones, such as DH32-29, DH33-27, U6, and DH201-2 [4,5,6]. In 2018, years of *Eucalyptus* plantation in China led to a coverage area of 5.4674 million ha [7]. *Eucalyptus* is frequently used as a raw material for charcoal, boards, pulp, energy, and furniture, due to its rapid growth, large biomass, and good wood quality. It is also important as an industrial timber forest species. The role of *Eucalyptus* as a huge carbon sink and its water conservation effect have led to positive effects in terms of ecological protection [8,9]. *Eucalyptus* is also used for extracting essential oils, mainly 1,8-cineole and camphene. These oils are rich in polyphenols, flavonoids, quinones, terpenoids, alkaloids, and tannins, which give these oils antibacterial, antiviral, and insecticidal properties [10,11].

*Eucalyptus* has superior growth characteristics, is adaptable to specific environments, and has desirable wood properties, making it one of the world’s most popular hardwood species. The demand for *Eucalyptus* has gradually increased in recent years. The traditional breeding methods for *Eucalyptus*, however, have involved long breeding cycles and poor genetic stability. In addition, biological stresses, such as *Ralstonia solanacearum*, *Ganoderma* sp., *Botrytis cinerea*, and *Cimex* and *Ophelimus bipolaris* pests, as well as non-biological stresses, such as wind and cold damage, have hindered the large-scale development of *Eucalyptus*, as these stresses result in a lack of long-term fine strains [12,13]. Plant tissue culture technology has, fortunately, allowed for the overcoming of these limitations associated with traditional breeding methods, thereby emerging as a powerful auxiliary tool for plant breeding and genetic improvement programs. Plant tissue culture technology involves regenerating complete plants from a particular plant tissue or organ (such as hypocotyls, cotyledons, leaves, stem segments, single genetically modified/genome-edited cells). A further application of plant genetic transformation technology involves the integration of excellent exogenous target genes and other genetic material into plant genomes, resulting in plants with novel genetic traits. This approach facilitates the rapid development of superior traits, guiding the targeted breeding and improvement of eucalyptus. The establishment of a stable and efficient eucalyptus regeneration system and genetic transformation system is of great significance in exploring the regulatory functions of the key genes involved and for germplasm innovation. Several plants have been optimized and developed through plant tissue culture technology and genetic transformation to date. However, due to the recalcitrance of *Eucalyptus*, its genetic transformation has remained difficult to achieve so far. Problems such as low conversion efficiency and instability have been encountered, making it difficult to meet the needs of scientific research and large-scale production. Advancements in theoretical and technical expertise, including those in the fields of genetic engineering, cell engineering, tissue culture, and somatic embryogenesis, have led to significant improvements in plant regeneration and genetic transformation. The methods of modern biotechnology allow for the overcoming of limitations of low regeneration efficiency and genetic transformation rates in most *Eucalyptus* species.

In this context, the present report summarizes the main research content and results that have been reported in the literature on the eucalyptus regeneration system and genetic transformation in recent years. In addition, the factors influencing *Eucalyptus* regeneration and genetic transformation are discussed to facilitate the construction of a stable and efficient *Eucalyptus* regeneration system and genetic transformation system by providing theoretical knowledge and technical support to accelerate the research on the molecular breeding of *Eucalyptus*.

## 2. Regeneration of *Eucalyptus*

### 2.1. The In Vitro Regeneration Methods of Eucalyptus

The regenerative ability of plants is applied widely in various horticultural and biotechnology programs and has its basis in pluripotency. The modalities used for plant regeneration include organogenesis, tissue repair, and somatic embryogenesis [14]. The regeneration methods used for *Eucalyptus* commonly include organogenesis and somatic embryogenesis [15] (Table 1). Organogenesis involves the direct development of injured or isolated tissues/organs into intact plants or the formation of calli, induced by PGRs. This method is also widely used to agricultural production [16,17]. Currently, the main regeneration method used for *Eucalyptus* organogenesis is “Shoot Proliferation”, which mainly involves plant regeneration by callus induction. Sussex et al. [18] pioneered using *E. camaldulensis* Dehnh. seedlings as explants to obtain calli in 1965. Subsequently, intact plants were induced through organ regeneration in *E. gunnii* Hook. f., *E. botryoides* Sm., *E. cinerea* F. Muell. ex Benth. and *E. maidenii*. Somatic embryogenesis is a unique process of plant development that involves producing asexual embryos from somatic cells or vegetative tissues without fertilization [19,20]. Asexual embryogenesis was first reported by Steward in 1958 in carrot cell suspension cultures [21]. Later, in 1981, Ouyang et al. [22] reported the first case of successful induction of embryonic cell mass and plant regeneration from the callus of *E. leichow* “*No:1*”. The eucalyptus species in which somatic embryogenesis has been conducted successfully include *E. citriodora* Hook., *E. grandis* W. Hill. ex Maiden, *E. nitens* Maiden, *E. globulus*, *E. tereticornis* Sm., and *E. maidenii* × *E. saligna* [23,24,25,26,27,28]. It has been reported that *Eucalyptus* might have two pathways for regeneration under different conditions [29]. For instance, Tan et al. [30] reported that, in *E. dunnii*, organogenesis could be achieved through axillary buds while regeneration could be achieved through somatic embryogenesis.

### 2.2. Factors Influencing the Regeneration of Eucalyptus In Vitro

#### 2.2.1. Explants

Explants are the initial source of contamination during inoculation, and, therefore, explant disinfection is a critical step in establishing a *Eucalyptus* regeneration system [43,44]. The commonly used surface sterilization methods for explant disinfection include ethanol, hydrogen peroxide, mercuric chloride (HgCl_2_), sodium hypochlorite (NaOCl), silver nitrate, and bromine water; for disinfection in *Eucalyptus* tissue culture, ethanol, HgCl_2_, and NaOCl are usually used [45]. The process of plant regeneration is affected by various factors, such as explant type, genotype, and culture conditions. The selection of suitable explants is the first step in plant tissue culture. As presented in Table 1, the explants used commonly for *Eucalyptus* culture include hypocotyls, cotyledons, stem segments, axillary buds, leaves, seeds, sprouts, and mature zygosperms, among other parts. Using stem segments as explants has reportedly produced a better bud induction effect, with an induction rate of over 60%. Azmi et al. [46] reported that the young and aged hypocotyls and cotyledons of *E. globulus* could not induce bud formation, with the buds regenerating only after 8–15 days. Fan et al. [47] reported that the highest rate of adventitious bud induction was achieved when the upper leaves (the 1st~4th leaves from the apex) were used as explants for *E. urophylla* × *E. tereticornis*. Genotype also has a great influence on *Eucalyptus* regeneration. Deepika et al. [48] evaluated the callus induction and bud regeneration ability of eight *Eucalyptus* genotypes and reported considerable differences among the 8 genotypes, with *E. grandis* (G1) and *E. grandis* × *E. nitens* (GN1) exhibiting the highest bud regeneration potential. Bandyopadhyay et al. [25] observed low-frequency somatic embryos on the surface of the *E. nitens* callus, while only tissue structures similar to somatic embryos were observed in the *E. globulus* callus. Corredoira et al. [28] used the shoot tips and leaves of *E. globulus* and *E. maidenii* × *E. saligna* hybrids as explants to induce somatic embryogenesis and reported higher embryogenesis in hybrid Sal-May.

#### 2.2.2. Culture Medium

In *Eucalyptus* regeneration culture, medium composition affects growth and development largely. Significant differences in bud induction and growth were noted among the different *Eucalyptus* varieties evaluated in different basic media in the present study (Table 1). The *Eucalyptus* callus culture and adventitious bud regeneration are usually conducted on the Murashige and Skoog (MS) basal medium. In certain specific kinds of *Eucalyptus*, basic media such as woody plant medium (WPM), JADS, EDMm, QLm, and B5 are also used. Oberschelp et al. [49] compared the effects of MS, WPM, JADS, and EDMm basal media on the micropropagation and nutritional status of four *E. dunnii* clones, reporting that the evaluated basal media could achieve axillary bud propagation, while the EDMm medium led to only bud growth with no greening or oxidation symptoms and a high rooting rate. de Oliveira et al. [50] attempted to reduce the effect of oxidation on the regeneration efficiency of *E. grandis* × *E. urophylla* by seeding the leaves with petioles into WPM1, MS, JADS, and QLm media, reporting the highest adventitious bud regeneration rate and the lowest oxidation rate in the WPM medium. In another study, the basal shoots of *E. benthamii* Maiden & Cambage trunk were evaluated in WPM, MS, and JADS media, and no significant difference in the amount of bud dry matter formed in the two subcultures was noted among the different media, although the WPM medium led to no oxidation [51].

#### 2.2.3. Plant Growth Regulators (PGRs)

Plant regeneration involves an intricate participation of PGRs, which are closely related to the plant species, concentration, and ratio. Auxin and cytokinin play pivotal roles in the regeneration process of *Eucalyptus*. When explants are incubated on a medium supplemented with auxin and cytokinin, *Eucalyptus* regenerates into seedlings in vitro. The commonly used PGRs include α-naphthalene acetic acid (NAA), indole-3-acetic acid (IAA), indole-3-butryic acid (IBA), 2,4-Dichlorophenoxyacetic acid (2,4-D), 6-furfurylaminopurine (kinetin), 6-Benzylaminopurine (BAP or BA), and thiazuron (TDZ) [52]. In the in vitro regeneration of *E. camaldulensis*, cytokinin BA (concentration 0.8–1.5 mg/L) and KT (concentration 0.3–1.0 mg/L) could effectively promote adventitious bud induction. After adventitious bud induction is completed, the concentration of cytokinin should be reduced in a timely manner to reduce its inhibitory effect on adventitious bud elongation [31]. Fernando et al. [53] reported that the leaves of three genotypes of *E. polybractea* R. T. Baker exhibited higher shoot regeneration rates on the BA-rich media. Wang et al. [54] indicated that while adding 2,4-D to the culture medium is more effective in inducing the callus tissue, it inhibits the regeneration of adventitious buds. This phenomenon is attributed to the fact that 2,4-D mainly promotes cell division and growth while having a relatively poor effect on organ formation [55]. TDZ plays a crucial role in bud formation, callus tissue induction, and somatic embryo development. Different concentrations of TDZ reportedly induced embryogenic callus tissue in *E. urophylla* × *E. grandi*s, although with significant differences in size and induction rate [56]. Brassinosteroids (BRs) are involved in the regulation of various processes of plant growth and development, as well as in several other important processes, such as responses to biotic and abiotic stresses [57,58]. Lower concentrations of BRs reportedly improve the induction rate of eucalyptus adventitious roots, while high concentrations of BR increase the number of cambium cells at the base of the stem, enhance the degree of lignification at the base, and inhibit the induction rate of adventitious roots [59]. The addition of GA3 to the culture medium could promote the formation of somatic embryos in the *Eucalyptus* callus tissue, while the addition of ABA inhibited germination and the direct entry into the late stage of somatic embryo maturation [60].

Different species of *Eucalyptus* have been adapted to different types and concentrations of PGRs for tissue regeneration (Table 2). The combination of NAA and 6-BA hormone is used frequently for adventitious bud induction in *Eucalyptus*, with the corresponding concentration ranges of 0.1~2 mg/L and 0.1~5 mg/L. However, Hajari et al. [61] reported that the use of the BAP+IAA combination of plant hormones was more suitable for the induction of callus regeneration in *E. grandis* × *E. urophylla* compared to the BAP+NAA combination. This observation was attributed to the effects of different genotypes. In a study on embryo induction in *Eucalyptus*, embryoid bodies were obtained by using stem segments from *Eucalyptus* tails on a medium supplemented with 0.1 mg/L NAA + 0.01 mg/L TDZ [62]. Nugent et al. [63] studied the somatic embryogenesis of *E. globulus* and reported that the cotyledons and hypocotyls could induce the development of somatic embryos in the medium supplemented with 50 µM Picloram + 100 µM IBA, although with a low frequency of occurrence and the obtained somatic embryos exhibiting bipolar and dysplasic cotyledons, shoots, and root tips. Ouyang et al. [64] reported that the hypocotyls of *E. pellita* performed the best effect in callus induction with 0.005 mg/L TDZ + 0.1 mg/L NAA, while the 1.0 mg/L 6-BA + 0.1 mg/L NAA treatment group led to the highest incidence of lower body embryos and a better growth status. The above results demonstrate that PGRs exert a great impact on *Eucalyptus* regeneration induction.

#### 2.2.4. Other Influencing Factors

In addition to explants, culture media, and PGRs, factors affecting the regeneration process for *Eucalyptus* include culture conditions such as light, temperature, and additives. The yellow and blue spectra are reported to be the most suitable for the in vitro propagation of *E. grandis* × *E. urophylla* without impairing bud development, suggesting that spectral quality affects explant development [68]. In a study, the stem segments of *E. urophylla* × *E. grandis* were treated with 5% sucrose solution at 4 °C for 12 h, which resulted in stronger embryonic callus activity and an induction rate of embryonic callus higher than that noted under dark culture conditions [69]. *Eucalyptus* is rich in phenolic substances, which are then exuded at the explant nodes on the medium, resulting in browning, which affects the transformation effect. The addition of anti-browning agents effectively inhibits the occurrence of this browning. Examples include the addition of 500 mg/L PVP to prevent the browning of explants when using the *E. tereticornis* callus to regenerate buds or the addition of 8 mg/L VC + 15 mg/L VB2 to the initial medium for the ex vivo regeneration of *E. saligna* × *E. exserta* [70]. It has been reported that the addition of the anti-browning agent VC reduces the rate of browning, although this also leads to a decreased rate of somatic embryo induction and might even cause complete inhibition of somatic embryogenesis [71,72]. Somatic variation significantly affects the regeneration ability of plants. Studies have reported that adding 500 mg/L casein hydrolysate and 500 mg/L glutamine can promote the development of somatic embryos in *E. citriodora*. However, in experiments with *E. globulus*, adding the same concentration of casein hydrolysate and glutamine did not increase the development of somatic embryos, but instead increased the frequency of abnormal somatic embryos [23,26]. In addition, conducting the dark culture treatment in the early stage of explant regeneration culture reportedly promotes adventitious bud differentiation, delays callus aging and browning, reduces explant necrosis, and effectively decreases the browning rate [65,73,74,75]. Fan et al. [76] reported that antibiotics and herbicides also affect the in vitro regeneration of *Eucalyptus*. When these authors incorporated different concentrations of Kanamycin, Hygromycin, Glyphosate, and Glufosinate–ammonium during the regeneration process of *E. grandis* × *E. urophylla* SP7, adventitious bud induction was completely inhibited at concentrations of 40, 10, 40, and 2.5 mg/L, respectively.

## 3. Genetic Transformation of *Eucalyptus*

Plant genetic transformation involves the application of recombinant DNA technology, cell tissue culture technology, or germplasm system transformation technology for the stable introduction of novel genes into the target plant genome. In order to improve the quality and yield of *Eucalyptus* breeding programs, and to increase tolerance to biotic and abiotic stresses, genetic transformation may be adopted to obtain plant varieties with excellent traits, such as high yield, high resistance, and high quality. Two types of genetic transformation may be applied: direct transformation and indirect transformation [77]. Direct genetic transformation involves the transfer of the gene of interest using external forces, such as particle bombardment, electroporation, microinjection, and pollen tube mediation, while indirect genetic transformation involves the transfer of the gene of interest into the target cell via *Agrobacterium* [78,79]. *Eucalyptus* species present great difficulty in genetic transformation, and several strategies have been used for their genetic transformation, including electroporation, particle bombardment, and *Agrobacterium-*mediated approaches. In 1992, Manders et al. [80] determined the electroporation parameters for the optimal expression of the *CAT* gene in the protoplasts of *E. citriodora*; namely, adjusting the voltage and pulse duration, increasing the plasmid concentration, and adding vector DNA. In 1996, Serrano et al. [81] used the gene gun method for the stable transformation of the mature zygotic embryos of *E. globulus*. Subsequently, Sartoretto et al. [82] used the gene gun method to transform the hypocotyls and cotyledons of *E. grandis* × *E. urophylla* and thereby obtained a GUS-positive callus, although the resulting callus did not exhibit regenerated transgenic buds.

The *Agrobacterium*-mediated approach is one of the most commonly used methods of transformation for *Eucalyptus* owing to its simplicity, high efficiency, good reproducibility, and low copy number. Dibax et al. [83] utilized the *P5CSF129A* gene for the in vitro regeneration and transformation of *Eucalyptus* trees. The *P5CSF129A* and *uidA* genes were integrated into the cotyledon explants of *E. saligna* Sm. via *Agrobacterium tumefaciens* EHA105. The subsequent PCR detection and southern blotting analyses confirmed the existence of these plant genome transfer genes, with the proline content in the transformed plant leaves being four times higher than that in the non-transformed plant. Navarro et al. [84] transferred the cold-inducible transcription gene *CBF* isolated from *E. gunnii* into *E. urophylla* × *E. grandis* using the *Agrobacterium*-mediated method and reported improved freezing tolerance, a reduced cell size, a smaller leaf area, and slow growth in the transgenic lines. Ouyang et al. [85] transferred the *aiiA* gene into the hypocotyls of *E. urophylla* × *E. grandis* via *Agrobacterium* and reported that the adequately expressed aiiA protein conferred significantly enhanced resistance to *Ralstonia solanacearum* compared to the non-transgenic plants. Wu et al. [86] investigated the effect of *EguGA20ox1* overexpression in *E. grandis* × *E. urophylla* on its lateral root development and reported significantly accelerated hairy root growth in the seedlings inoculated with *EguGA20ox1*. Klocko et al. [87] generated three vectors overexpressing the *Arabidopsis thaliana FT* gene (*AtFT*) and introduced them into the leaves of *E. grandis × urophylla* hybrid (SP7) via *Agrobacterium tumefaciens*-mediated transformation. The results demonstrated that overexpression of the *AtFT* gene can significantly accelerate flowering in *Eucalyptus*, with normal development of flowering structures leading to viable seed and pollen production. Following this, Nagle et al. [88] employed CRISPR/Cas9 technology to target and knockout three reproductive-related genes in *Eucalyptus*, subsequently these knockout events were then transformed into SP7 clones using an *Agrobacterium-mediated* transformation process. The study demonstrated that the ablation of *ETDF1*, *EREC8*, and *EHEC3*-*like* leads to male sterility or, in some cases, complete sterility in *Eucalyptus*. Numerous studies, to date, have reported the establishment of *Agrobacterium*-mediated genetic transformation systems for *Eucalyptus*, such as *E. urophylla* × *E. grandis*, *E. grandis* × *E. urophylla*, *E. camaldulensis*, *E. grandis*, *E. tereticornis* clones, *E. saligna*, and *E. urophylla* × *E. tereticornis* (Table 3).

### 3.1. Genotype

In the process of *Agrobacterium*-mediated genetic transformation of *Eucalyptus*, several factors affect the transformation efficiency, such as genotype, *Agrobacterium* strains, bacterial concentration, and culture time [95]. Different *Agrobacterium* genotypes exhibit significantly different transformation efficiencies [96]. After the *Agrobacterium*-mediated inoculation and kanamycin culture, the hypocotyls without shoot apex in *E. globulus* exhibited signs of rapid necrosis, while the hypocotyls with shoot tips remained active [43]. Plasencia et al. [97] used *Agrobacterium rhizogenes* A4RS to infect the hypocotyls, radicle apex, and stem segments of 14-day-old *E. grandis* seedlings, each kind separately, and reported that the hypocotyls led to the best transformation efficiency. Aggarwal et al. [90] evaluated the transformation efficiency of three clones (“T1”, “CE2”, and “Y8”) of *E. tereticornis* and reported significant differences in transformation efficiency among the different clones. Zhou et al. [98] used 12 *Eucalyptus* species (26 genotypes) to conduct regeneration ability experiments and reported good regeneration ability for these genotypes. Afterward, the hypocotyls and stem segments of seven genotypes with a high regeneration ability for *Agrobacterium*-mediated genetic transformation were evaluated, which revealed that RO1 and CA1 were more sensitive to *Agrobacterium*, and the transformation efficiency of the stem segments was just half of that observed for the hypocotyl.

### 3.2. Agrobacterium Species and Bacterial Liquid Concentration

Different *Agrobacterium* strains exhibit differences in their ability to successfully transfer T-DNA to various *Eucalyptus* species. The selection of suitable transformation strains is, therefore, important for plant transgenic works. The *Agrobacterium* species commonly used in the genetic transformation of *Eucalyptus* are *A. tumefacien*s and *A. rhizogenes*. In the genetic transformation of *Eucalyptus* seedlings, Moralejo et al. [99] evaluated the ability of three different strains of *A. tumefaciens*—namely C58C1 pMP90, EHA101 pEHA101, and LBA4404 pAL4404—to infect the *Eucalyptus* tissues. The fluorescence measurement results reflecting transient GUS expression 6 days after infection indicated that the transient expression levels of EHA101 pEHA101 were four times the levels of LBA4404 pAL4404 and twice the levels of C58C1 pMP90. Machado et al. [100] reported significant differences in the susceptibility of five *A. rhizogenes* and twelve *A. tumefaciens* wild-type strains to the genetic transformation of *Eucalyptus* trees, with *E. grandis* × *E. urophylla* presenting high sensitivity. Wang et al. [94,101] found that strain GV3101 had the best and highest transformation efficiency when infecting the stem internodes of *E. urophylla* × *E. tereticornis*, while strain EHA105 had the highest transformation efficiency when infecting the leaves of *E. urophylla* × *E. grandis*. When the leaves of *E. urophylla* × *E. grandis* were infected with *A. rhizogenes* MSU440, a conversion rate of 20.2% was determined using PCR molecular detection and GUS staining [102]. The optimal state of *Agrobacterium* is generally determined using the visible-light optical density value (optical density at 600 nm, OD_600nm_) to indicate the optimal concentration of the *Agrobacterium* used. Ahad et al. [103] genetically transformed *E. camaldulensis*, achieving the non-precultured bacterial suspension OD_600nm_ of 0.5 and the highest transformation efficiency ever achieved using the bacterial suspension (OD_600nm_ = 0.3). Aggarwal et al. [90] reported high transient GUS expression in the leaf explants of *E. tereticornis* at the EHA105 solution concentration of 0.8 OD_600nm_.

### 3.3. Antibiotics

Antibiotics are essential factors affecting *Agrobacterium*-mediated genetic transformation and may be divided into two categories: (1) antibiotics utilized to screen transformants, such as kanamycin, G418, hygromycin, and spectinomycin; and (2) antibiotics that inhibit the growth of *Agrobacterium*, such as cephalosporin, cefotaxime sodium, ceftriaxone sodium, and carbenicillin. Kanamycin is the earliest screening agent that is currently being used in plant genetic engineering. Lai et al. [104] reported that a kanamycin concentration of 75 mg/L led to the almost complete inhibition of the callus differentiation of the stem segment of *E. urophylla* × *E. grandis*, while the differentiation of adventitious buds was completely inhibited. When the concentration of kanamycin was increased further, the stem segment exhibited browning and even died, which was not conducive to explant regeneration. Southerton also reported that kanamycin, hygromycin, and tetracycline concentrations above 50 mg/L, 1 mg/L, and 5 mg/L, respectively, inhibited the regeneration rate of *E. occidentalis* Endl. [105]. Oliveira et al. [106] treated the cotyledons of *E. saligna* with three different concentrations of kanamycin and reported that lower concentrations of kanamycin were more favorable to bud induction. de França Bettencourt et al. [41] improved the transformation efficiency of *E. urophylla* BRS07-01 by using 50 mg/L kanamycin as the screening agent. Wang et al. [94] reported the use of 15 mg/L kanamycin for screening resistant adventitious buds in *E. urophylla* × *E. grandis*. The above results indicated that lower concentrations of kanamycin are more suited to screen for *Eucalyptus* genetic transformations. Adventitious bud regeneration and root formation are highly sensitive to hygromycin, with 5 mg/L hygromycin leading to an almost complete inhibition of adventitious bud regeneration and root formation, as reported by Wang et al. [94] for *E. urophylla* × *E. grandis* DH32-29. Labate et al. [107] used 15 mg/L hygromycin as the screening agent for the transformation of *E. camaldulensis*, *E. grandis*, and *E. urophylla* × *E. grandis*.

### 3.4. Preculture and Co-Culture

Preculture time is one of the important parameters affecting the genetic transformation of plants. A certain period of explant preculture prior to *Agrobacterium* infection reportedly promotes cell division, facilitates exogenous DNA integration, improves tolerance to *Agrobacterium*, reduces the destruction of *Agrobacterium* explants during the transformation process, and improves the efficiency of genetic transformation. Prakash and Gurumurthi reported that the preculture of the cotyledons and hypocotyls of *E. tereticornis* for two days followed by a co-culture with *Agrobacterium* reduced allergic reactions and improved transformation efficiency [108]. Wang et al. [109] performed variance analysis (ANOVA) and reported that co-culture time, infection time, optical density of the bacterial solution, pre-incubation time, and the pH of the infection solution exerted significant effects on the differentiation of the adventitious buds of *E. urophylla* × *E. grandis*, with the effect of the co-culture being the largest. Spokevicius et al. [110] co-cultured the stem segments infected with the *Agrobacterium* strain AGL1 for 2, 3, 4, 5, 6, and 7 days followed by GUS staining; they reported the highest transient GUS expression rate after 6 days of co-culture. When the leaves of *E. grandis* × *E. urophylla* were infected with *Agrobacterium* followed by a preculture for 2 days and then a co-culture for 3 days, the efficiency of *uidA* gene expression was at its highest [111].

### 3.5. Other Factors

The genetic transformation efficiency of *Eucalyptus* is also influenced by factors such as additives, experimental methods, and operations. When Thanananta et al. [93] increased the contact area between the explant and *Agrobacterium* by peeling off the epidermal layer around the axillary buds of *E. camaldulensis* × *E. tereticornis*, the conversion rate was improved. da Silva et al. [112] reported that the addition of 100 µM acetosyringone (AS) to the medium during the co-culture stage of *E. saligna* bud tips promoted the transient expression of the *uidA* gene and reduced the toxic effects of kanamycin. For instance, the transient GUS expression index was not significantly different when AS was added to the bacterial solution prior to the genetic transformation of the *E. grandis* clone Eg5, which led to the poor staining of explants, ultimately causing significant alterations in the transformation of *Agrobacterium.* On the other hand, the transient expression index reached the best value of 150 mg/L when AS was added during the co-culture stage [113]. However, in the genetic transformation of *E. grandis* × *E. urophylla*, the transformation efficiency was higher when AS was not added to the co-culture medium [114], which could be attributed to large differences in the sensitivity of different *Eucalyptus* varieties to AS. WUS protein is involved in the regulation of meristem development and regeneration, and the addition of truncated peptides isolated from the WUS protein to the co-culture medium reportedly improves the callus transformation efficiency of *E. grandis* × *E. urophylla* DH32-29 [115]. González et al. [116] used *A. tumefaciens* and achieved an average conversion rate of 9.16% without sonication and 17.64% after 30 s of sonication, indicating that sonication also affects the conversion rate of *Agrobacterium*.

## 4. Several *Eucalyptus* Regeneration and Transformation Workflow

### 4.1. Genetic Transformation of E. urophylla × E. grandis DH32-29 Leaves [94]

Using *neomycin phosphotransferase II* (*nptII*) as a selection marker gene and *β-glucuronidase* (*uidA*) as a reporter gene, seven transformed plants were ultimately obtained through GUS staining and PCR detection, with a transformation rate of approximately 1.9%. The specific experimental process was as follows:

(1) Explant: the in vitro leaves from microshoots were subcultured for 20–25 days of *E. urophylla* × *E. grandis* DH32-29.

(2) Pre-culture: the leaves were precultured in liquid WPM basal medium supplemented with 0.02 mg/L NAA and 0.24 mg/L CPPU for 7 days.

(3) *Agrobacterium*-mediated transformation: the leaves were infected in a suspension of *A*. *tumefaciens* strain EHA105 (OD600 = 0.3) for 30 min.

(4) Co-culture: the infected leaves were co-cultured in liquid WPM basal medium supplemented with 0.02 mg/L NAA and 0.24 mg/L CPPU without antibiotics, under dark conditions at 25 ± 2 °C for 72 h.

(5) Callus induction: the leaves were cultured in a liquid WPM medium supplemented with 0.02 mg/L NAA and 0.24 mg/L CPPU containing 100 mg/L cefotaxime (Cef), 100 mg/L timentin (Tmt), and 15 mg/L kanamycin (Kan) for 15 days to induce callus formation.

(6) Adventitious bud induction: the calluses were transferred to a solid WPM medium supplemented with 0.10 mg/L NAA, 0.50 mg/L 6-BA containing 100 mg/L Cef, 100 mg/L Tmt, and 15 mg/L Kan, and the fresh medium was replaced every 15 days until resistant adventitious buds appeared.

(7) Rooting culture: the resistant adventitious buds were transferred to a rooting medium 1/2MS supplemented with 0.10 mg/L NAA containing 200 mg/L Cef and 75 mg/L Kan to induce the formation of adventitious roots.

(8) Transplantation: the verified transgenic shoots were planted in the soil and grown in a greenhouse.

### 4.2. Genetic Transformation of E. urophylla × E. grandis DH32-29 Hypocotyls [85]

Using the *hygromycin phosphate transferase* (*HPT*) gene as a selection marker, an *aiiA* plant expression vector Pcam-PPP3-*aiiA* was constructed for genetic transformation. The positive rate of fluorescence screening for positive transformation plants was 40%, the induction rate of resistant buds was 64.36%, and the rooting rate was 76%. The specific experimental process was as follows:

(1) Explant: hypocotyls were grown from stem segments of aseptic clonal seedlings *E. urophylla* × *E. grandis* DH32-29.

(2) Pre-culture: the hypocotyls were pre-cultured on MS medium supplemented with 13.2 μM PBU and 0.285 μM IAA for 6 days.

(3) *Agrobacterium* transformation: the hypocotyls were incubated in a suspension of *Agrobacterium* tumefaciens strain EHA105 (OD600 = 0.3) in the dark for 2 h.

(4) Co-culture: the transformed hypocotyls were co-cultured on MS medium supplemented with 13.2 μM PBU and 0.285 μM IAA for 2 days.

(5) Callus induction: callus formation was induced on MS medium supplemented with 13.2 μM PBU and 0.285 μM IAA.

(6) Adventitious bud induction: transformed plants were selected on MS medium supplemented with 200 mg/L Cef and 9 mg/L hygromycin (Hyg); 0.25 μM BA and 4.4 μM NAA was used to select the transformed resistant buds.

(7) Rooting culture: Hyg-resistant buds were transferred to 1/2MS medium supplemented with 2.46 μM IBA for rooting culture.

(8) Transplantation: after rooting, the plants were transferred to a greenhouse for acclimatization and were eventually transplanted into soil.

### 4.3. Genetic Transformation of E. urophylla × E. grandis DH32-29 Stem Segments [109]

The mCherry gene was fused with the 35S promoter and recombined with the pHDE-Cas9 plasmid to construct the pHDE/Cas9-35S-*mCherry* plasmid. The positive rate of fluorescence identification was about 42%. The specific experimental process was as follows:

(1) Explant: stem segments were grown from aseptic clonal seedlings *E. urophylla* × *E. grandis* DH32-29.

(2) Pre-culture: the stem segments were pre-cultured on MS medium supplemented with 13.2 μM PBU and 0.285 μM IAA for 6 days.

(3) *Agrobacterium*-mediated transformation: the stem segments were infected with *Agrobacterium* tumefaciens strain EHA105 (OD600 = 0.3, pH = 5.8) for 2 h.

(4) Co-culture: the infected stem segments were co-cultured on MS medium supplemented with 13.2 μM PBU and 0.285 μM IAA for 2 days.

(5) Callus induction: callus induction was performed on MS medium supplemented with 13.2 μM PBU and 0.285 μM IAA.

(6) Adventitious bud induction: the calluses were transferred to MS medium supplemented with 200 mg/L Cef, 0.25 μM BA, and 4.4 μM NAA for adventitious bud induction, with the medium replaced every 7–10 days.

(7) Adventitious bud elongation: the induced adventitious buds were transferred to 1/2MS medium supplemented with 6.6 μM PBU and 0.285 μM IAA to promote bud elongation.

(8) Induction of adventitious roots: adventitious buds elongated to 2–3 cm were transferred to 1/2MS medium supplemented with 2.46 μM IBA for adventitious root induction.

(9) Transplantation: when the seedlings had grown 5–10 adventitious roots, approximately 5 cm in length, they were acclimatized and transplanted.

(10) Fluorescence screening: fluorescence microscopy was used to screen for positive transformed plants carrying the *mCherry* gene.

(11) Molecular identification: PCR technology was used to identify the Cas9 gene in positive transformed plants to validate the transformation results.

### 4.4. Genetic Transformation of E. saligna Cotyledons [91]

Using the *gus* gene as a reporter gene, the transformation efficiency was determined to be 1.5% through *gus* gene expression detection as well as through PCR analysis of the transformed plants. The specific experimental process was as follows:

(1) Explant: cotyledons were grown from 12 day *E. saligna* seedlings.

(2) Pre-culture: pre-culture did not have a significant effect on transformation efficiency.

(3) *Agrobacterium* transformation: the cotyledons were infected with a suspension of *Agrobacterium* tumefaciens strain EHA105 for 30 min.

(4) Co-culture: the infected cotyledons were co-cultured on MS medium supplemented with 50 µM AS, 4.4 µM BAP, and 2.7 µM NAA for 5 days.

(5) Adventitious bud regeneration: the cotyledons were cultured on MS medium supplemented with 12.5 mg/L kanamycin, 4.4 µM BAP, and 2.7 µM NAA for 180 days for the selection and regeneration of resistant adventitious buds, with fresh medium replaced every 15 days.

## 5. Conclusions

*Eucalyptus* exhibits rapid growth and strong adaptability, which has led to its widescale application. The development of *Eucalyptus* regeneration and genetic transformation systems is of great significance to the genetic improvement of *Eucalyptus*. The key to success in *Eucalyptus* genetic improvement is the establishment of relevant tissue culture regeneration systems and genetic transformation systems. However, few studies have reported the regeneration and genetic transformation of Eucalyptus trees so far. The regeneration and genetic transformation of *Eucalyptus* depend on various factors, such as explants, plant growth regulators, culture media, *Agrobacterium* species and concentrations, culture time, antibiotics, and light. Different *Eucalyptus* varieties require different conditions for regeneration and genetic transformation. Therefore, it is imperative to develop an efficient and stable *Eucalyptus* genetic transformation and regeneration system to achieve the breeding goals of strong resistance, a short breeding cycle, and low production costs. The establishment and genetic transformation of the *Eucalyptus* regeneration system has progressed considerably, although certain challenges remain in the following aspects: (1) the establishment of a unified regeneration scheme suitable for most varieties of *Eucalyptus* due to the various options available for the selection of explant materials and application of PGRs, along with the effects of light, oxide, and other factors; (2) the *Agrobacterium*-mediated approach, which is the most commonly adopted genetic transformation method, presents an unstable transformation frequency and a low conversion rate, limiting large-scale production; and (3) molecular studies on eucalyptus began quite late, but 44 Eucalyptus genomes have been sequenced and many important genes have been obtained. Due to the immature genetic transformation system, the genes cannot be verified for their functions in eucalyptus. Therefore, the molecular research related to eucalyptus wood property and resistance is still in its infancy. Future research could, therefore, focus on the following aspects: (1) establishing an efficient and stable *Eucalyptus* regeneration system and transformation system to improve transformation efficiency; (2) conducting technological research and development further conducive to the genetic transformation of *Eucalyptus* and accelerating the research on important functional genes; (3) accelerating the application of gene editing technology to the field of *Eucalyptus* genetic transformation to resolve the concerns of genotype dependence and low transformation efficiency.

## Figures and Tables

**Table 1 plants-13-02843-t001:** The established *Eucalyptus* regeneration systems.

Species	Regeneration Method	Explant Type	Culture Medium	References
*E. globulus*	Organogenesis	Hypocotyl	MS	[25]
*E. camaldulensis*	Organogenesis	Leaves	MS	[31]
*E. dunnii*	Organogenesis	Stem segments	MS	[32]
*E. urophylla* S. T. Blake	Organogenesis	Hypocotyl	SPCa	[33]
*E. urophylla* × *E. grandis*	Organogenesis	Stem segments	MS	[34]
*E. urophylla* × *E. grandis* (DH32-29)	Organogenesis	Whitened internodal stems	B5	[35]
*E. pellita* F. Muell.	Organogenesis	Stem segments	MS	[36]
*Corymbia torelliana* F. Muell.	Organogenesis	Stem segments	MS	[37]
*E. urophylla × E. grandis*	Organogenesis	Stem segment with auxiliary buds	MS	[38]
*E. grandis × E. camaldulensis*	Organogenesis	Stem segments	MS	[39]
*E. nitens* (H.Deane & Maiden) Maiden	Organogenesis	Hypocotyl	MS	[25,26,27,28,29,30,31,32,33,34,35,36,37,38,39,40]
*E. urophylla* (BRS07-01)	Organogenesis	Leaves	WPM	[41]
*E. cloeziana*	Organogenesis	Cotyledonary	MS	[42]
*E.* “*leichow*” “*No:1*”	Somatic embryogenesis	Seeds	B5/H	[22]
*E. citriodora*	Somatic embryogenesis	Cotyledonary	B5	[23]
*E. grandis*	Somatic embryogenesis	Leaves	MS	[24]
*E. globulus*	Somatic embryogenesis	Mature zygotic embryos	MS	[26]
*E. tereticornis*	Somatic embryogenesis	Mature zygotic embryos	MS	[27]

**Table 2 plants-13-02843-t002:** Treatments involving the addition of growth regulators for adventitious bud induction and rooting cultures in certain kinds of *Eucalyptus*.

Species	Differentiation Pathways	Induce Differentiation	Bud Regeneration	Rooting Culture	Reference
*E.* “*leichow*” “*No:1*”	Somatic embryogenesis	0.8–1.0 mg/L KT + 2.0 mg/L 2,4-D	0.2–1.0 mg/L BA + 0.5 mg/L NAA	0.5 mg/L NAA/0.5–1.0 mg/L IBA	[22]
*E. globulus*	Somatic embryogenesis	3–15 mg/L NAA			[26]
*E. nitens*	Callus induction	1.0 mg/L NAA + 0.5 mg/L BAP	0.5 mg/L NAA + 1 mg/L BAP	3 mg/L IBA	[25]
*E. camaldulensis*	Adventitious bud induction	0.8 mg/L BA + 0.3 mg/L KT + 0.05 mg/L NAA	0.8 mg/L BA + 0.05 mg/L TDZ + 0.1 mg/L NAA	0.5 mg/L NAA	[31]
*E. dunnii*	Adventitious bud induction	0.5 mg/L 6-BA	0.5 mg/L 6-BA + 0.1 mg/L NAA and 0.3 mg/L 6-BA + 0.2 mg/L NAA	0.5 mg/L IBA	[32]
*E. urophylla*	Callus induction	0.57 µM IAA + 6.6 µM PBU	0.54 µM NAA + 0.44 µM BA + 0.3 µM GA3	2.5 µM IBA	[33]
*E. urophylla* × *E. grandis*	Callus induction	13.2 µM PBU + 0.285 µM IAA	6.6 µM PBU + 0.285 µM IAA	2.46 µM IBA	[34]
*E. urophylla* × *E. grandis* (DH32-29)	Callus induction	0.02 mg/L NAA + 0.6 mg/L TDZ and 0.02 mg/L NAA + 0.6 mg/L CPPU	0.5 mg/L BA + 0.1 mg/L NAA	0.5 mg/L IBA	[35]
*E. pellita*	Callus induction	2.0 mg/L PBU + 0.05 mg/L IAA	0.8 mg/L 6-BA + 0.1 mg/L NAA	0.5 mg/L NAA + 0.5 mg/L IBA	[36]
*Corymbia torelliana*	Adventitious bud induction	0.5 mg/L 6-BA + 0.1 mg/L NAA	0.2 mg/L 6-BA + 0.1 mg/L NAA and 0.3 mg/L 6-BA + 0.2 mg/L NAA	0.2 mg/L IBA + 0.6 mg/L ABT	[37]
*E. urophylla* × *E. grandis*	Callus induction	1~2 mg/L 2,4-D	0.5 mg/L 6-BA	2.0 mg/L NAA	[38]
*E. grandis × E. camaldulensis*	Adventitious bud induction	0.2 mg/L NAA + 0.5 mg/L 6-BA	0.15 mg/L NAA + 0.5 mg/L 6-BA	0.15 mg/L NAA + 1.0 mg/L IBA	[39]
*E. cloeziana*	Callus induction	0.05 µM TDZ	0.05 µM TDZ + 4.44 µM BAP	0.89 µM BAP + 0.05 µM NAA	[42]
*E. grandis* × *E. urophylla*	Callus induction	0.1 µM NAA + 0.25 µM TDZ	0.5 µM NAA + 5.0 µM BA	2.46 µM IBA	[50]
*E. grandis*	Callus induction	0.25 mg/L NAA + 0.12 mg/L TDZ	0.3 mg/L IAA + 0.3 mg/L 6-BA + 0.05 mg/L IBA	0.4 mg/L NAA	[65]
*E. camaldulensis*	Adventitious bud induction	0.1 mg/L NAA + 2 mg/L BAP	0.5 mg/L BAP	1.0 mg/L IBA	[66]
*E. bosistoana*	Callus induction	1 mg/L NAA + 1.5 mg/L BA	0.5 mg/L NAA + 1.5 mg/L BA	1.0 mg/L NAA	[67]

**Table 3 plants-13-02843-t003:** *Eucalyptus* species for which a genetic transformation system has been preliminarily established.

Species	Gene	Explants	Transformation Type	Transformation Situation	Reference
*E. citriodora*	*CAT*	Cotyledonous protoplasts	electroporation	47%	[80]
*E. globulus*	*GUS* and *nptII*	Mature zygotic embryos	particle bombardment	Successful of transformation	[81]
*E. urophylla* × *E. grandis*	*aiiA*	Hypocotyl	*Agrobacterium-*mediated	PCR:40%	[85]
*E. camaldulensis*	*GUS*	Hypocotyl	*Agrobacterium-*mediated	Successful of transformation	[89]
*E. tereticornis* clones CE2	*uidA* and *nptII*	Leaves	*Agrobacterium-*mediated	Successful of transformation	[90]
*E. saligna*	*GUS*	Cotiledonary	*Agrobacterium-*mediated	1.5%	[91]
*E. urophylla*	*Rs-AFP2*	Hypocotyl	*Agrobacterium-*mediated	PCR:7%	[92]
*E. globulus*	*Coda* (*GUS)*	Hypocotyl	*Agrobacterium-*mediated	Successful of transformation	[43]
*E. camaldulensis* × *E. tereticornis*	*HAL2*	Axillary bud	*Agrobacterium-*mediated	PCR:24%	[93]
*E. urophylla* × *E. grandis* DH32-29	*uidA* and *nptII*	Leaves	*Agrobacterium-*mediated	1.9%	[94]

## Data Availability

The original contributions presented in the study are included in the article. Further inquiries can be directed to the corresponding author.

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
