# Peer review of "Regeneration and Genetic Transformation in Eucalyptus Species, Current Research and Future Perspectives"

_plants, 2024, doi:10.3390/plants13202843_

Round 1

Reviewer 1 Report

Comments and Suggestions for Authors

The review article deals with the regeneration of plants of the genus Eucalyptus using tissue culture techniques, and with the genetic transformation methods. First, the authors listed the main methods for plant regeneration (different pathways of organogenesis, somatic embryogenesis) and then discussed factors influencing the regeneration of Eucalyptus plants in vitro (types of explants, culture medium, PGRs and other factors). In the second part, the authors summarised direct and indirect genetic transformation methods and factors influencing transformation success such as plant genotype, Agrobacterium species and strains, use of antibiotics, preculturing (explant preculture prior to Agrobacterium inoculation) and co-culturing, and other factors.

Since type of explant, culture medium, plant genotype and various other factors influence transformation and regeneration success, it would be interesting to subdivide the chapters by type of explant and see how successful the regeneration was, which PGRs were used by the explant, which regeneration pathway was triggered, etc. Now each chapter includes information for different types of explant.

Also, the whole workflow (from tissue preparation for transformation to plant regeneration) could be analysed by explant type, as the medium and PGRs can have an influence from one phase to another (e.g. PGRs used for shoot multiplication can still have a great impact on rooting, etc.). This structure would be more in line with one of the aims – to create a standardised regeneration scheme suitable for most Eucalyptus species. In this way, it would be easier for readers to find out which explants to use, how to prepare tissue cultures for Agrobacterium inoculation or other transformation techniques, and which PGRs and media should be used for plant regeneration. Perhaps an additional chapter could summarise the best practices/most commonly used workflows that could serve as suggestion for an efficient and stable Eucalyptus regeneration and transformation system.

It is noted that cotyledons and hypocotyls are among the most commonly used as explants, but other types of explants from clonally propagated species should be better addressed since frequently the aim is often to preserve the genotype of the selected clone.

Here is an example of a study reporting on an efficient system for regeneration and transformation of adventitious buds in Eucalyptus (and which is also cited by the authors, but only once in the table). Could more useful information be obtained from the latest studies?

https://www.frontiersin.org/journals/plant-science/articles/10.3389/fpls.2022.1011245/full

For example, leaves were used here https://onlinelibrary.wiley.com/doi/10.1111/pbi.12431

The same method was used here for knocking out floral and meiosis genes using CRISPR/Cas9 (https://onlinelibrary.wiley.com/doi/full/10.1002/pld3.507).

I didn’t observe anything related to the somaclonal variability, which could also be an important factor in determining the regeneration protocol.

The objective of the review was to provide a reference for solving the issues of genetic instability(*) and low transformation efficiency of Eucalyptus and to construct an efficient and stable Eucalyptus regeneration and transformation system to accelerate the process of the genetic improvement of Eucalyptus. I think that the last objective (establishment of an efficient transformation/regeneration system) could be addressed better, as I already mentioned, with an additional chapter summarising all most promising practices.

(*What did the authors mean by genetic instability? Silenced transgene, difficulties with construct insertion into the genome, etc.?)

But this is just my opinion and I would be happy to hear the authors' views on this.

I also have some comments on specific lines:

2. Page

64: …leaves, stem segments, etc. – But also from a single genetically modified/genome-edited cell.

72: Several plants have been optimized and developed through plant tissue culture technology and genetic transformation to date. - I would suggest including some recent examples of traits or some other not so recently introduced important traits using transgenesis and genome editing techniques.

96: … and production – what is meant by this word?

page 10: polypeptides isolated ... – original article mentioned truncated peptides from WUS specific domain

401-403: molecular research on Eucalyptus began quite late – the NCBI datasets contain 44 Eucalyptus genomes, so it looks like there is quite a lot of genetic information available for Eucalyptus species?

Reviewer 2 Report

Comments and Suggestions for Authors

The article is a review paper to my understanding, so please indicate that to the editorial office.

 The article discuss about eucalyptus species in the context of breeding so I would change also the title. My proposal is "Regeneration and genetic transformation in eucalyptus species, current research and future perspectives"

Comments on the Quality of English Language

Line 12: rewrite the sentence “and genetic improvement of important genes in this tree” and “possible genetic breeding capacity of important genes of the species”

Line 14: rephrase the sentence “of Eucalyptus genetics and breeding” with “of Eucalyptus genetics and breeding programs”

Line 15: replace “report” with “review”

Line 17: erase “on each link”

Lines 19-24

Rewrite as: “Furthermore the study also summarises the problems encountered in Eucalyptus regeneration and genetic transformation, with reference to previous studies, and outlines future developments and prospects. The aim was to provide a reference for solving the problems of genetic instability and low transformation efficiency of eucalyptus, and to establish an efficient and stable eucalyptus regeneration and transformation system to accelerate the process of its genetic improvement.

Line 30: Eucalyptus “is”mainly originated

Line 33: erase “regions” in the sentence low tropical regions

Line 45: replace “owning” with “due to”

Lines 49-51: rewrite the sentence as “These oils are rich in polyphenols, flavonoids, quinones, terpenoids, alkaloids and tannins, which give these oils antibacterial, antiviral and insecticidal activity.”

Lines 52-54: rewrite as “Eucalyptus has superior growth, adaptability to specific environments and desirable wood properties, making it one of the world's most popular hardwood species.”

Line 60: erase “for”

Line 66-68: rewrite as: This approach facilitates the rapid development of superior traits, guiding the targeted breeding and improvement of eucalyptus.

Line 73-75: rewrite as “Problems such as low conversion efficiency and instability are encountered, making it difficult to meet the needs of scientific research and large-scale production.”

Line 80: change paragraph

Line 80: change “contents” with “content”

Line 68,80 etc: Eucalyptus must be changed in the text to “eucalyptus” because you refer to common name and not the Latin name so there is no need to put it in Capital the first letter and to italics.

Line 94-95: change to “also widely used to agricultural production”

Line 234: erase “in addition to increasing” and replace with “and to increase”

Line 295: Too long sentence place a dot after “transgenic works.” And initiate the next sentence “The Agrobacterium species….”

Line 322: initiate new sentence “The kanamycin is the ….)

Line 350: what is OD? Write the whole name.
